

# Perspectives on the misconception of levitating soil aggregates

Gina Garland[1,2], John Koestel[1,3], Alice Johannes[1], Olivier Heller[1], Sebastian Doetterl[2], Dani Or[2,4] & Thomas Keller[1,3]

[1]Agroscope, Department of Agroecology & Environment, Reckenholzstrasse 191, CH-8046 Zürich, Switzerland
[2]Swiss Federal Institute of Technology ETH, Department of Environmental Systems Science, Universitätstrasse 16, CH-8092 Zürich, Switzerland
[3]Swedish University of Agricultural Sciences, Department of Soil & Environment, Box 7014, SE-75007 Uppsala, Sweden
[4]Desert Research Institute, Division of Hydrologic Sciences, 2215 Raggio Parkway, Reno, NV, 89512, USA

*Correspondence to*: Gina Garland (gina.garland@usys.ethz.ch) and Thomas Keller (thomas.keller@agroscope.admin.ch)

**Abstract.** Soil aggregation is an important process in nearly all soil and land-use types across the globe. Aggregates develop over time through a series of abiotic and biotic processes and interactions, including plant growth and decay, microbial activity, plant and microbial exudation, bioturbation, and physicochemical stabilization processes, and are greatly influenced by soil management practices. Together, and through feedbacks with organic matter and primary soil particles, these processes form dynamic soil aggregates and pore spaces, which together constitute a soil's structure and contribute to overall soil functioning. Yet, the concept of soil aggregates is hotly debated, leading to confusion about their function or even existence. We argue here that the opposition to the concept of soil aggregation likely stems from the fact that the methods for characterization of soil aggregates have largely been developed in the context of arable soils, where tillage promotes the formation of discrete soil aggregates that are easily visible in the topsoil. We propose that the widespread use of conceptual figures showing detached and isolated aggregates can be misleading and has contributed to the skepticism towards the validity or relevance of studies on soil aggregates. However, the fact that we do not always see distinct aggregates within soils *in-situ* does not mean that aggregates do not exist. Here, we illustrate how aggregates can form and dissipate within the context of undisturbed, intact soils, highlighting the point that aggregates do not necessarily need to have a distinct physical boundary and can exist seamlessly embedded in the soil. We hope our contribution helps to alleviate the debate on soil aggregates and supports the foundation of a shared understanding on the characterization and function of the 'dual nature' of soil structure.

## 1 Introduction

Soil structure is defined as "the spatial heterogeneity of the different components or properties of soil" (Dexter, 1988). In particular, the organization of these particles into solids, aggregates, voids, and pore networks largely determines the capacity of soils to retain and transmit water, oxygen, and various other organic and inorganic substances through the soil profile (Bronick and Lal, 2005; Rabot et al., 2018). This structure not only provides habitat for soil organisms – and is in turn influenced by their activities – but the interaction between the physicochemical soil environment and its biological communities drives numerous environmental processes including root growth and plant development, nutrient cycling and carbon sequestration, water purification, and protection against erosion (Lal, 1991; Sullivan et al., 2021). Together, these



functions play a vital role in the provision of soil ecosystem services, thus further highlighting the importance of soils for directly contributing to a multitude of sustainability goals (Lehmann et al., 2020; Lal et al., 2021).

Yet while the importance of soil structure as a foundation for sustaining key soil functions is increasingly recognized, there exists no unified technique or single metric to characterize the structure of a soil. Rather, multiple approaches have been developed, each targeted towards a particular research question or aspect of soil structure (Rabot et al., 2018; Vogel et al., 2021; Yudina and Kuzyakov, 2023). For example, the establishment of non-invasive imaging methods as a means to directly quantify and visualize structural properties, such as pore networks, in undisturbed soil is progressing rapidly and promises to

become a standard in future soil research (Rabot et al., 2018; Schlüter et al., 2020). In addition, another frequently measured indicator of soil structure is aggregate stability. While this term has come to mean different things depending on the context and spatial scale of research (Amézketa, 1999), it generally refers to the degree to which a soil remains aggregated under various physical, chemical, biological and environmental stresses (Tisdall and Oades, 1982; Kemper and Rosenau, 1986; Six et al., 2000a; Papadopoulos, 2011). Soil aggregates, in turn, are broadly defined as two or more primary soil particles that

cohere more strongly to each other than neighboring particles (Martin et al., 1955; Kemper and Chepil, 1965; SSSA, 1997).

Soil aggregation occurs at multiple spatial scales and is driven by a variety of complex and dynamic biotic and abiotic interactions. The scale at which soil aggregation occurs, coupled with the specific mechanism(s) binding soil particles together, directly impacts the strength of these soil bonds, and thus overall aggregate stability (Yudina and Kuzyakov, 2023). A vital

driver of soil aggregation is related to the proportion and types of iron and aluminum oxides and clay minerals in a given soil, as well as organic matter either applied externally or derived from plants and soil organisms, which forms organo-mineral complexes with clay particles and is integral for soil carbon sequestration (Tisdall and Oades, 1982; Hemmingway et al., 2017; Totsche et al., 2018). The degree to which aggregation occurs is simultaneously driven by multiple abiotic processes including flocculation and cementation of clay particles, as well as shrinking-swelling processes induced by changes in soil moisture

and temperate (Bronick and Lal, 2005; Totsche et al., 2018; Pihlap et al., 2021). Together with bioturbation by macrofauna (Wilkinson et al., 2009; Piron et al., 2017) and activity of microorganisms and growing plant roots (Rillig and Mummey, 2006; Lehmann et al., 2017), these dynamic processes create both soil aggregates and soil pores, both of which are important aspects of soil structure and regulators of soil functioning.

Yet despite the long-standing acknowledgement of both soil pore spaces (Rabot et al., 2018; Vogel et al., 2021) and soil aggregates (Emerson, 1959; Edwards and Bremner, 1967; Chenu et al., 1998), some researchers have questioned whether aggregates exist at all. This doubt was first introduced over three decades ago (Letey, 1991), and has continued until now with recent debates on the function of aggregates (Kravchenko et al., 2019; Wang et al., 2019; Yudina and Kuzyakov, 2019). One of the main critiques of using aggregates to characterize soil structure and assess soil functioning is the inherent destruction of

the soil required for such assessments (Young et al., 2001), the fact that aggregate properties depend on the method used to



isolate them (Letey, 1991), as well as the unrealistic boundary conditions of isolated aggregates (Kravchenko et al., 2019; Vogel et al., 2021). Furthermore, it has been claimed that it is not possible to identify soil aggregates in X-ray images of consolidated undisturbed soil, or at least not in the same size and proportion as the soil aggregates measured from destructive measurement techniques (Baveye, 2020). Albeit only very few attempts have been made to corroborate or falsify this claim

(e.g. Koestel et al., 2021), it has been misconstrued as "proof" that aggregates do not exist.

This strong opposition between viewpoints on soil structure has led to a rift in the soil science community, essentially dividing researchers into two groups. For despite the well-accepted definition of soil structure which integrates both the solids and pore spaces (Dexter, 1988), in practice what we see today is one group focusing primarily on aggregates (the *solid phase*

*perspective*) and the other on the pore network morphology (the *pore space perspective*), with very little overlap (Rabot et al., 2018; Vogel et al. 2021; Yudina and Kuzyakov, 2023). We believe that this scientific divide is not only unnecessary, but is in fact hindering the progression of research in the field of soil science. In an effort to reduce the confusion surrounding these contradictory aspects of soil structure and to bring a foundation of shared understanding in the soil science community, here we discuss and illustrate how aggregates do not necessarily need to have a distinct physical boundary to exist in the soil profile.

We do not attempt to choose a side in the "solid phase perspective" versus "pore space perspective" debate, as we believe there is in fact no contradiction between these concepts for describing soil structure. Rather, we aim to demonstrate that there is no incongruity between the existence of aggregates and the fact that we often cannot see them in undisturbed soil, which we hope helps resolve some of the conflicting views, and ultimately advances our understanding of soil functioning.

## 2 Aggregates do not require a distinct physical boundary

We believe that part of the controversy and confusion surrounding soil aggregates is rooted in conceptual models that display detached, isolated aggregates which seem to levitate (Fig. 1), while in practice aggregates are often not visible in undisturbed soils or in deeper soil layers at the same spatial scale (Fig. 2). This likely stems from research on soil structure emerging from the study of tilled, arable soils (Dexter, 1988; Elliott and Coleman, 1988; Or et al., 2021), where soil aggregates are indeed discrete units that are easily visible in the topsoil layers (Fig. 2a). Here we argue that soil aggregates do indeed exist, but do

not necessarily look like these classic images of soil aggregates seen in drawings and arable fields (Figs. 1 and 2a). While this distinction may be obvious for many in the soil science fields, there is apparently some confusion, wherein the simplistic, conceptual images created to highlight the mechanistic process of aggregate formation and disintegration is taken as a realistic depiction of soil aggregates. Here we address this and show that while this may be true in certain topsoils, this is rarely the case in undisturbed and deeper soil layers. In fact, one of the oldest and most widely used definitions of soil aggregates describe

them as "any group of soil particles that coheres more strongly to each other than neighboring particles" (Martin et al., 1955; SSSA, 1997). Given this understanding of soil aggregates, it is logical then that they may, but do not necessarily need to be, bordered by pore space at spatial scales relevant for most soil mechanistic investigations. Our viewpoint here challenges the concept of intra- versus inter-aggregate pore space: if aggregates do not need to be physically separated, there is not necessarily



a distinctive inter-aggregate pore space. Instead, as soil aggregates within intact soils do not levitate, they logically must be in
physical contact at a minimum of one point, and thus we argue that aggregate boundaries are rather defined by planes or regions
of weaker cohesion.

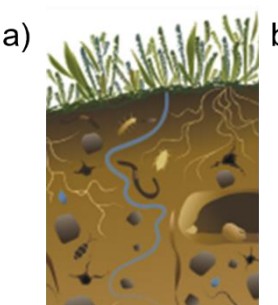 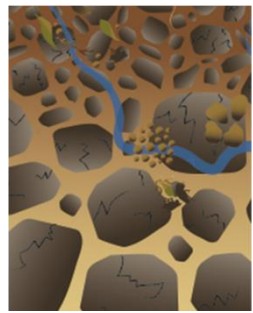

**Figure 1. Conceptualization of aggregates that are detached from each other and isolated within the soil** yields confusion
about aggregate boundaries and how they are embedded in soil. This illustrative example from FAO (2015) shows detached
and isolated aggregates (dark brown) from a) topsoils and b) deeper soil horizons.

To illustrate this idea conceptually, we first show an example of two soil aggregates each consisting of three soil particles (Fig.
3). For illustration purposes, we represent the different soil particle fractions (i.e. sand, silt and clay) as simple single unit-
sized squares, with the outer edges of each square representing one or more of the various biotic and abiotic binding agents
(i.e. microbial or plant-derived polysaccharides, electrostatic interactions between clay particles, mycorrhizal fungi, etc.). Here
we do not explicitly account for the nature of the binding agents and organic matter involved in the soil aggregation process,
as the specific binding agents and mechanisms happen simultaneously and dynamically, and differ depending on the local
(micro-)climatic conditions, soil mineralogy and texture, biological components and the scale at which aggregates are assessed
(μm to m). Furthermore, these processes have been described in detail and are not the focus of our discussion (Totsche et al.,
2018; Yudina et al., 2018; Yudina and Kuzyakov, 2023). We instead highlight the relative strength of these binding agents
between aggregate constituents, whereby the lines connecting two squares represents bond strength between soil particles at a
given period of time, with thicker lines indicating a higher bond strength (Fig. 3). In this example, we show that two separate
soil aggregates can exist adjacent to each other without interaggregate pore space when there is either a) a weaker inter-
aggregate bond compared to the intra-aggregate bonds or b) no binding force between adjacent aggregates if we assume that
these are pressed together (i.e. confined) by the surrounding soil structure (Fig. 2b,c and Fig. 3). Pore space between
neighboring soil aggregates can of course also occur, in cases when there is no binding force between adjacent aggregates (Fig.
3c), as evidenced for example by soil aggregates formed by tillage which can be seen with the unaided eye (Fig. 2a), or at
smaller spatial scales where advanced imaging techniques are necessary to visualize this pore space (i.e. Lucas, 2021).
However, it goes without saying that soil aggregates embedded in the soil, both with or without visible pore space, must still
be in physical contact at a minimum of one point with surrounding soil particles.





**Figure 2. Aggregates are clearly visible following tillage in topsoils but are indistinguishable at deeper soil horizons and over time.** a) Aggregates are clearly evident in a freshly tilled soil (photo by Dani Or, ETH Zürich, Switzerland) but are not visible in b) deeper soil layers of undisturbed soils at the same spatial scale. c) The temporal evolution of soil structure after tillage based on X-ray computed tomography images (visible pores >120 µm, based on voxel size of 60 µm): aggregates are clearly visible directly after tillage (left panel, June 2018) but coalesce and fuse with time (right panel, October 2020) (Koestel, unpublished).




As we have discussed, the term 'soil aggregate' is used to indicate that certain soil particles cohere to each other more strongly than neighboring particles. It does not give any indication of the size, shape, strength, or general arrangement of the particles and voids that make up that aggregate. Instead, the size and composition of individual soil aggregates depend on the technique used and force applied to separate them. Over the past decades, numerous methods have been developed to investigate and

categorize soil aggregates, including wet-sieving, dry-sieving, drop-shatter tests, laser diffraction, sedimentation, and visual assessments (Yudina et al., 2018; Yudina and Kuzyakov, 2019). For example, microaggregates (<250 µm in diameter) are shown to consist of organic matter strongly stabilized via mineral associations that are relatively long lasting and thus contribute more prominently to carbon sequestration, while macroaggregates (>250 µm in diameter) typically consist of more labile particulate organic matter pools, and thus are more strongly linked with microbial and plant community dynamics

(Blanco-Canqui and Lal, 2004; Lavallee et al., 2019). Together, the various processes guiding soil aggregation have a profound impact on a multitude of soil characteristics including soil porosity (Fukumasu et al., 2021) and the turnover and stabilization of soil carbon pools (Weng et al., 2021), which in turn ultimately contributes to overall soil structure.

a) Weak binding force between aggregates; no pore space

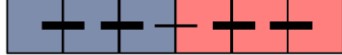

b) No binding force between aggregates; no pore space



c) No binding force between aggregates; pore space

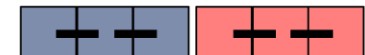


**Figure 3. Aggregates do not need to be separated by distinct physical boundaries.** Simplified conceptual illustration showing possible arrangements between two individual soil aggregates (indicated in blue and red). Soil particles are represented by squares, and black horizontal lines indicate bonds between soil particles, with bond strength indicated by line

thickness. The three cases shown are examples of aggregates which: a) share a weak binding force between adjacent aggregates and therefore show no visible pore space, b) have no binding force between adjacent aggregates but still show no visible pore space, and c) have no binding force between aggregates and are physically distinct from each other, resulting in visible pore space between them.




Given this conceptual description of a soil aggregate, we now illustrate how they may form and subsequently dissipate within intact soils using the growth of a plant as an example (Fig. 4). We do not aim to model real processes, but instead to show how aggregates may appear and disappear as the binding strengths between soil aggregates and particles strengthen and weaken over time. To simplify our point, we start with a soil structure consisting of a heterogeneous mixture of aggregates (formed as

described in Fig. 3), unaggregated primary soil particles and organic matter, and pore spaces prior to plant growth ($t_0$). As the plant roots develop and elongate through the soil profile, the existing aggregates and unaggregated soil particles become bound to each other and neighboring particles through a combination of abiotic and biotic processes as described above ($t_1$).  As the plant continues to grow, assimilating additional carbon-rich substances derived from the atmosphere via photosynthesis into the soil, this additional organic matter as well as the increased microbial activity surrounding these organic matter hot spots

create further bonds between soil particles ($t_2$). Additionally, changes in temperature and moisture patterns, along with oxygen availability at microsites during plant growth impact various abiotic processes (e.g. shrinking-swelling and interactions of carbonates, clays, and iron and aluminum oxides), thus further influencing aggregation either directly or through interactive effects with biological processes. Over time, this activity can either form new aggregates, or combine two or more aggregates into larger aggregates, depending on the relative bond strengths between the soil particles. Once the plant completes its

lifecycle and ultimately dies, its tissues are decomposed by soil microorganisms. As these more labile organic compounds are consumed and eventually depleted, the biological complexes and bonds between soil particles become weaker, and many of the relatively weakly bound aggregates (i.e. macroaggregates) eventually disintegrate (Tisdall and Oades, 1982; Oades, 1984; Six et al., 2004) ($t_3$). However, the ease and speed at which these aggregates disintegrate, or 'turnover' as it is often described (Six et al., 2000b), is directly related to the size and the strength of the bonds between particles. For example, some aggregates

have relatively fast turnover times (between 30 and 88 days, DeGryz et al., 2005) while more strongly bound aggregates (i.e. microaggregates) have been shown to endure for decades up to centuries (Totsche et al., 2018; Yudina and Kuzyakov, 2023) ($t_4$). We note that disintegration does not necessarily mean physical separation, but instead that the bonds between soil particles of aggregates may simply weaken until they are no longer associated (see Fig. 4). Over time, any remaining aggregates will either dissipate completely, or in the case of the relatively more stable microaggregates, can become incorporated into newly

forming soil aggregates, where the cycle continues ($t_x$). It is important to note, however, that even in our simplified example of one main process of root growth, soil particles are aggregated together heterogeneously and via numerous and simultaneously acting mechanisms, with distinct differences in aggregate size and stability depending on the specific combination of mechanisms involved (Yudina and Kuzyakov, 2023).




time

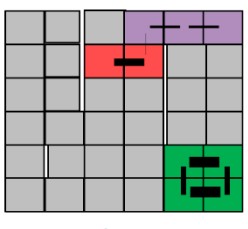

$t_0$: For simplicity, we start with a mixture of aggregates (formed as in Fig. 3, indicated with colored squares), unaggregated soil particles (gray squares), and pore spaces.

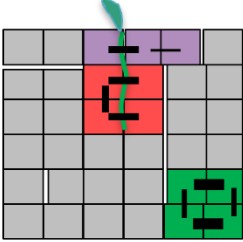

$t_1$: A plant grows, forming additional aggregates and pores around the root by binding existing aggregates and unaggregated particle via various biotic and abiotic processes.

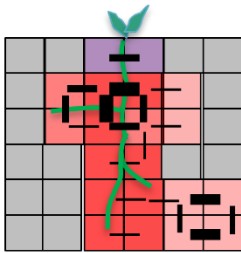

$t_2$: As the roots continue to grow, additional organic inputs enter the soil and biological activity is stimulated. Even stronger bonds are formed between soil particles and microaggregates through organo-mineral complexes.

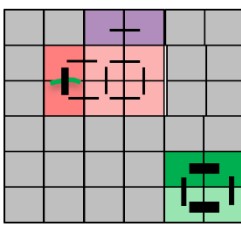

$t_3$: After the plant dies, the roots begin to decompose. As biological activity wanes, the complexes between soil particles weakens, whereupon some of the aggregates begin to dissipate.

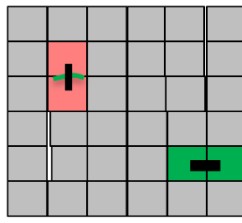

$t_4$: Eventually, only the relatively more strongly bound aggregates remain.

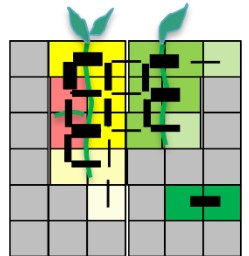

$t_x$: Over time, these aggregates can either completely weaken into unaggregated soils, or become incorporated into newly forming aggregates.



**Figure 4. Formation and dissipation of soil aggregates embedded in soil.** In this conceptual illustrative example of *in-situ* aggregate formation, unaggregated soil particles, existing soil aggregates (formed as in Figure 3), and pore spaces are represented as individual grey squares, and the outer edges of each square represent one or more of the various different biotic and abiotic binding agents (i.e. microbial or plant-derived polysaccharides, electrostatic interactions between clay particles). The lines represent bonds connecting neighboring particles, with the relative strength of the bond indicated by the thickness of the line. The strength of bonds in this example are arbitrarily attributed but still mimic a realistic scenario of how mineral particles and their abiotically and biotically-derived bonds and their decomposition result in the growth and disintegration of aggregates embedded in soil, without considering the exact nature of bonds (i.e. our illustration does not aim at explicitly simulating mechanisms). Different colors indicate different aggregates.

While the above simplified illustrative conceptualization is not novel in terms of describing the general formation and turnover processes of soil aggregates during plant growth and decay, we highlight the fact that these processes can and, most often do, occur within intact soils, with aggregates seamlessly embedded in soil. The fact that aggregates are not always visible in intact soils via various imaging techniques exactly as in the classic representation of detached, isolated aggregates (i.e. Fig. 1, 2a) is on the one hand logical considering that they are surrounded by soil and must therefore be in physical contact to some degree (i.e. they are not levitating). On the other hand, we propose that there is also some confusion due to the issue of scale (see Fig. 4). Given the various strengths of the different binding mechanisms, as well as the fact that multiple processes occur simultaneously to aggregate soil at a given time, it is well-known that soil aggregates of different sizes exist. These typically range from micrometers up to several centimeters in diameter, depending on the technique used and force applied to separate them (Six et al., 2004; Nimmo, 2013; Yudina and Kuzyakov, 2019). Therefore, considering that aggregates have been described ranging in sizes of up to 10,000 times difference, they may vary in visibility from being easily seen with the unaided eye (Fig. 2a), visible only with microscopy (see Vidal et al., 2018) or X-ray imaging tools (Koestel et al., 2021; Fig. 2c), or not visible at all with current imaging techniques. Therefore, what we actually see in an image, and therefore can conclude regarding the arrangement of solids and voids, often greatly depends on the scale and resolution of a sample. For example, in an X-ray µCT image, the scale at which a soil is assessed greatly impacts the reported values of cumulative pore size distribution (Lucas, 2021). Moreover, if soil aggregates do not necessarily have to be bound pore space, as we propose here, then current imaging techniques would not be able to visually capture these aggregates regardless of the scale, since we cannot "see" the mechanical dimension of soil structure, i.e. the strength of the bonds between soil particles. Thus, to conclude that soil aggregates do not exist in intact soils simply because they are not visible at one particular spatial scale does not allow for a valid assessment of soil structure.





## 3 Implications and future research trajectories

Understanding the role of soil structure in driving various soil functions and ecosystem services has been at the forefront of research for decades and has led to many important findings and advancements in analytical technologies. Yet despite this

progress, much remains unknown regarding the dynamic interplay between the physical, chemical, and biological components, as well and the pore spaces and voids of a soil's structure and how they together drive soil functioning across different spatial scales. To echo Vogel et al. (2021), a holistic approach is indeed necessary to link the effects of soil structure on soil functioning. However, despite the numerous advancements in technology, there is not one single methodological approach that can provide a complete overview of the three-dimensional arrangement of soil pores, the composition and bioavailability

of nutrients contained in the solid mineral particles and organic matter, and the composition and activity of biological communities contained in a given space over time. Therefore, a more holistic assessment must inevitably consider multiple complementary approaches.

The limitations of assessing soil aggregates and their relation to soil functions using classic destructive analytical techniques

are clear and have been discussed in detail (Rabot et al., 2018; Kravchenko et al., 2019; Vogel et al., 2021). The question of how exactly an aggregate isolated from the soil in such a way corresponds to the very same aggregate *in-situ* remains indeed unresolved, and thus care should be taken regarding which insights we can derive from studies on aggregates and which not (Kravchenko et al., 2019). Yet despite methodological limitations, continuing to investigate the spatial arrangement of solid particles and voids as well as the flow and transfer of substances throughout the soil pore network and how the solid particles

are connected remains critical. The composition and bioavailability of substances bound in aggregates, and how this influences soil microbial communities and soil fauna living within or on the surface of aggregates and pores is paramount to better understand how soil structure drives functioning. With the understanding that soil aggregate formation and turnover does occur in intact soils, without the need for distinct physical boundaries, we hope that future research can unite these important aspects of soil structure with those of soil pore networks for a better representation of soil structure and the functions it provides.

## 4 Conclusions

The widespread use of conceptual figures showing detached and isolated aggregates is misleading and has largely contributed to the confusion about the function or even existence of soil aggregates. Based on the spatial scale investigated and the processes that contribute to their formation and turnover, it is clear that they can, but do not need to be, separated by distinct physical boundaries. The fact that we often do not see aggregates (e.g. in X-ray images) in undisturbed soils or deeper soil

layers with distinct pore boundaries comparable to those in topsoils of freshly tilled arable soils does not mean that aggregates do not exist, only that in most cases they are seamlessly embedded in the soil. Rather than furthering the divide between researchers in the opposing *pore space perspective* compared to the *solid phase perspective*, we support previous research emphasizing the vital point that aggregates and pore space are intimately linked and that both soil aggregation and soil pore



formation are important for furthering our understanding of soil structure dynamics. However, the question of how aggregates

that are seamlessly embedded in soil can be isolated and analyzed in more detail *in-situ* to more closely assess microscale

processes within aggregates and in relation to pore structures still remains a considerable challenge, yet a worthwhile and

crucial future research goal.

**Author contributions**

The conceptualization of the idea for this paper was done by GG, JK, AJ, OH, SD, DO, and TK. GG and TK wrote the initial

draft and all authors were involved in the review and editing of the paper.

**Declaration of Competing Interest**

The authors declare that they have no known competing financial interests or personal relationships that could have appeared
to influence the work reported in this paper.

**Acknowledgements**

This research was funded by Agroscope, ETH Zurich, and the Swiss National Science Foundation. All authors contributed to
the conceptualization, discussion, and writing of this manuscript.

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
