# Peer review of "Perspectives on the misconception of levitating soil aggregates"

_EGUsphere, 2023_

## Referee Comment (RC1)

Review on MS https://doi.org/10.5194/egusphere-2023-1144

**"Perspectives on the misconception of levitating soil aggregates"**

**submitted by Garland et al.**

In the paper provided, the authors set out to try on two conflicting views on soil structure – solid and pore perspectives. In my opinion, they almost succeeded, as they provided a simple and, most importantly, viable abstraction to describe the delineation of the different functional zones within soil structure (which are actually aggregates) (*Figure 3*). This is the most important and strongest takeaway of this paper. Also, despite the simplicity and brevity of Section 3, I would like to emphasize its value for those readers who may wish to develop the topic of soil structure. Especially, I would like to highlight the validity of the thesis "a more holistic assessment must inevitably consider multiple complementary approaches". I have some suggestions and questions regarding the terminology and the wholeness of the proposed approach, which do not lessen the value of proposed MS. The MS contains the necessary novelty, and is written quite logically. Therefore, it can be accepted after a moderate revision. See my general and specific suggestions below.

**General comments**

Since the authors set out to try on two groups of researchers, it would be nice to see explicitly which of the main theses presented by both sides the authors find useful, and some arguments regarding the non-contradiction (L81) (in a general sense, I agree with you). For example, the authors do not use or discuss the definitions given in previous articles (*Vogel et al., 2022; Yudina and Kuzyakov, 2023*). As Prof. Vogel correctly points out in the preprint comments "*the wording has a significant impact on our perception*".

You need to better disclose what you mean by "do not necessarily need to have a distinct physical boundary" (L21) without jeopardizing the use of the term "aggregates" (which is actually what Prof. Baveye immediately appeals to in his comment to the preprint). In my understanding, this is analogous to applying the concept of soil types at the landscape scale, and trying to identify corresponding to different soils areas. It is also often impossible to distinguish *distinct* physical boundaries between them (as well as between other environmental bodies characterized by gradual transitions). Because many phenomena in nature are continuous, they don't know how to have sharp boundaries. However, we nevertheless use such abstractions and more than successfully. Accordingly, I don't think the word " distinct" is quite appropriate here. Because the distinctness will be achieved due to the conditions that we will set (real, i.e. experimentally affecting the soil, or virtually, i.e. modeling the processes).

From L136-138, it is not entirely clear what the authors' position is – whether they consider size, shape, etc. characteristics important for aggregates. Moreover, from L138-139 it appears that the authors take an operational approach – aggregates are what is distinguished in such and such a way. This to some extent contradicts the authors' position that aggregates exist as natural entities (L89), and the aggregate size differs (L141-143, L209). Please clarify your point.

I think it will be important to emphasize that the delineation of boundaries between aggregates according to the presented conceptual scheme on *Figure 3* (L108-125) will 1) strongly depend on the available instrumental capabilities (which pores and type of contacts are detectable), 2) and, accordingly, have different significance depending on the type (hierarchy) of the aggregates that the task is to delineate. If we are talking about the experimental application of your approach (that looks promising). For example, at a rather good resolution of tomographic imaging (of the order of the first µm), we do not see pores, which can actually constitute a meaningful fraction (see Gerke et al., 2021). Accordingly, if one is trying to apply what the authors suggest to research, the question arises as to how it should be used. It would be good to reveal this point to a greater extent within the Section 3 of MS.

I find the argument against inter- and intra-aggregate pore space (L97-101) as interesting, but in need of refinement. It highlights the weak side of the paper – you do not explicitly discuss the hierarchy of soil structural organization. But actually discussing it (e.g., here L182-185). In my opinion, without this abstraction (as pores belonging to different dynamical parts of the structure), as well as without the concept of non-aggregated mass (which you do not use exept L165 during describing t0 and beyond), presented approach does not look quite complete. Likely such abstractions would have improved the description in the L161-188.

**Specific comments**

L9 it's not very correct to write "soils and land-use types" with "and", because they are different kinds of things; land use type has no aggregation, only soils

L27 what is the difference between solids and aggregates? voids and pores? – It's not clear what the point of listing alternative terms here is. Please, try to use terms more strictly within the whole text of the MS.

L55 temperate → temperature

L66 "the unrealistic boundary conditions of isolated aggregates" conditions for what, modelling? or boundary conditions for isolation of aggregates?

L70 probably, the relevant works where this is said should be cited at the end of this sentence

L95-97 it will be good to put this argument to the abstract, as this is one of the important foundations on which your proposal is built; however, it is not clear for me what you mean here – "at spatial scales relevant for most soil mechanistic investigations". Please, clarify. E.g., microaggregates having size of several-tens-hundreds μm and surrounded by micro- and mesopores are relevant for rheological behavior.

L98 "if aggregates do not need to be physically separated" → prefer to write "cannot be surrounded on all sides by pores". Because you actually make an excellent point that the physical boundary can be not only the pore, but also various types of contacts between solids.

L109 "the different soil particle fractions (i.e. sand, silt and clay)" → the term "soil particle fractions" is not strict enough – as for the soil texture fractions it would be more accurate to write "soil particle size fractions". If you mean them. According to the bracketed text, you do. However, it would be more logical and consistent with the rest of the text to specify simply «soil solids» instead of this term referring to soil texture rather than soil structure. Please, clarify this.

L138 "Instead … " – I cannot agree with this sentence in any way, because if aggregates are natural formations of soil, their size and other characteristics are determined by soil processes, but not by methods used.

L152 "Aggregates do not need to be separated by distinct physical boundaries." would be better named –> "Types of physical boundaries between the aggregates". Pores and different types of contacts between soil solids are physical boundaries. Our task as researchers is to define correctly the boundary conditions for different types of aggregates.
* * *
Gerke, K. M., Korostilev, E. V., Romanenko, K. A., & Karsanina, M. V. (2021). Going submicron in the precise analysis of soil structure: A FIB-SEM imaging study at nanoscale. *Geoderma*, *383*, 114739.

---

## Author Response (AR1)

**CC1:**

In their manuscript submitted to EGUsphere, in many ways like Yudina and Kuzyakov (2017, 2023) before them, Garland et al. (2023) attempt to "support the foundation of a shared understanding on the characterization and function of the 'dual nature' of soil." In essence, these authors all try to reaffirm the central importance of aggregates in soil science, and argue that the two current schools of thought about soil structure, one explicitly focused on aggregates and the other not involving them at all, are not fundamentally at odds with each other but are rather complementary and "two sides of the same coin". As Yudina and Kuzyakov (2019) put it, "a dual pore-aggregate perspective is necessary for holistic understanding of soil structure and functions". Garland et al. (2023) seem to consider that by emphasizing this complementarity between the two conflicting approaches, they would succeed to "alleviate the debate on soil aggregates". One could contend that, thanks to their text, this latter goal is now largely fulfilled, however not quite in the way these authors seem to have hoped. Indeed, since they manage to provide very strong and convincing arguments against the relevance of the concept of aggregates to the description of processes occurring in soils, I feel that the fate of aggregates in that context should now logically be considered sealed.

Thank you for your opinion. Perhaps you mean with your summarization "the fate of the aggregates in that context should now logically be considered sealed" that the difference between "fraggregates" (see comment by Vogel below and our associated response) and "aggregates" has now firmly been established. In this case, yes, we agree. "Fraggregates" do not form naturally and seamlessly in the soil, whereas aggregates, defined as "any group of soil particles that coheres more strongly to each other than neighboring particles" (Martin et al., 1955; SSSA, 1997), and mentioned at L21, L99, and elsewhere throughout the text, do indeed.

To understand fully why that is, it is important first to grasp what the major differences are between the two opposing schools of thought and therefore what the alleged "dual nature" of soils could be. At one point in their manuscript, Garland et al. (2023) refer to the two approaches as the "solid phase perspective" and the "pore space perspective", respectively, but one could argue that these expressions are largely misleading, can easily be misunderstood, and therefore should be avoided at all cost, as was done, e.g., by Vogel et al. (2022) and Baveye et al. (2022). Probably far better labels would be the "aggregate perspective" and the "architectural perspective".

While all of these expressions, and more, have been used in previous work on this topic, we do not think one is inherently better than another at getting our point across. We initially chose not to use 'aggregate perspective' versus 'architectural perspective' because this implies that the aggregate perspective does not consider the architecture of the soil to be important, which is certainly not the case. However, in the updated manuscript we included both terminologies for each perspective (i.e. '*solid phase*' and '*aggregate*' perspective as well as '*pore space*' and '*architecture*' perspective) (L78-79

and L218-219) in order to include a broader range of existing expressions for these differing perspectives.

Both perspectives acknowledge that, as Garland et al. (2023) write in the first sentence of their abstract, "soil aggregation is an important process in nearly all soils and land-use types across the globe". However, whereas the "aggregate perspective" postulates that aggregation leads by definition to the formation of distinct aggregates, and views these aggregates as essential to understand the dynamics of soil structure and soil processes, the "architectural perspective" considers instead that the geometrical configuration of the solid phase and of the pore space in their undisturbed state, without necessarily requiring any reference to aggregates, is key to the description of processes occurring in soils.

I think you are misinterpreting our statement regarding the 'aggregate perspective'. In L93-98 we state "Here we argue that soil aggregates do indeed exist, but do not necessarily look like these classic images of soil aggregates seen in drawings and found in arable fields (Figs. 1 and 2a). While this distinction may be obvious for many in the soil science fields, there is apparently some confusion, wherein the simplistic, conceptual images created to highlight the mechanistic process of aggregate formation and disintegration is taken as a realistic depiction of soil aggregates. Here we address this and show that while this may be true in certain topsoils, this is rarely the case in undisturbed and deeper soil layers." With this we mean that the 'aggregate perspective' simply highlights the fact that within a soil profile, some "groups of soil particles cohere more strongly to each other than to neighboring particles" (Martin et al., 1955; SSSA, 1997). This leads to a heterogeneous arrangement of organo-mineral complexes in the soil matrix that will undoubtedly have an impact not only on the soil structure, but also the distribution of chemical and biological materials.

As we emphasize in the manuscript, however, these pockets of more or less strongly bound soil particles do NOT typically lead to distinct aggregates visible in the soil profile (i.e. "fraggregates", see comment from Vogel below). In fact, we specified (L127-128): "It does not give any indication of the size, shape, strength, or general arrangement of the particles and voids that make up that aggregate." Additionally, in the revised manuscript we made this point clearer by stating (L72-74): "Albeit only very few attempts have been made to corroborate or falsify this claim (e.g. Koestel et al., 2021), here we argue that such *in-situ* identification is not necessary, and in many cases is not realistic, for verifying the existence of soil aggregates".

The foremost point of divergence between the two schools of thought is therefore not about the existence of aggregates in soils. As reviewed in detail in Baveye et al. (2022), there are clearly a variety of different "natural aggregates" in soils, originating from a range of biotic or abiotic processes, and it would not make much sense to deny that. It would be ludicrous, for example, to ignore the fact that highly weathered soils in Central America, are able to withstand rainfalls that total more than 12 m annually because the clay particles and oxides that constitute them almost entirely are tightly bound in the form of "pseudo-sand" aggregates, so that their hydrology is that of sandy soils (e.g., Radulovich et al., 1992). In many of these situations where natural aggregates undeniably exist, very minimal disturbance of

the soils enables these aggregates to be identified. In other cases, much more significant disturbance is needed to isolate chunks of soil that can then be considered to be aggregates.

Thank you for your comment. In the revised manuscript we removed the statement that the existence of aggregates is questioned, as also brought up by Reviewer #2. Instead, we state that "some researchers have questioned the relevance of aggregates for soil processes" (64-65) (i.e. not the existence of aggregates themselves). In addition, we have highlighted that aggregates naturally created in the soil are not the same as artificially isolated aggregates (i.e. "fraggregates") by including the following sentence (L104-106): "As a result, aggregates formed *in-situ* will inherently not look the same as destructively isolated aggregates, but rather appear seamlessly embedded in the heterogeneous organo-mineral soil matrix, punctuated at various points by pore spaces, as described by Vogel et al. (2021)." Moreover, naturally formed soil aggregates do not require any disturbance (not even "very minimal") to create, because the process of binding more strongly to some neighboring soil particles than other is the formation itself, which is completely separate from the isolation method.

Where there is a fundamental disagreement between the two schools of thought, however, is about whether or not it is necessary to take aggregates explicitly into account when trying to understand the effect of the structure/architecture of soils on soil functions, and especially when trying to describe or predict soil functions quantitatively. These questions have been the object of much debate for at least 30 years (e.g., Letey, 1991; Amézketa, 1999; Young et al., 2001) and one could say even since the early work of Redlich (1940) and Russell (1941, 1971) more than 80 years ago (Baveye et al., 2022). From the start, scientists faced the challenges of measuring the geometrical characteristics of aggregates and of knowing how to use the resulting information in quantitative models. Historically, in order to determine which types of aggregates are present in a given soil, and what their shape, composition, and size distribution are, two routes have been followed, both equally disruptive. The first consists of air-drying soil samples, impregnating them with resin, slicing them, and observing aggregates in the resulting thin sections (e.g., FitzPatrick, 1993). Aside from possible artefacts associated with the preparation of the thin sections, a key disadvantage of this approach is that it can only provide a snapshot of aggregates at one instant of time, which prevents any consideration of their dynamics. The second approach consists of dismantling soils, by imparting them increasing levels of energy, so as to break them down into progressively smaller fragments. Again, this approach can provide information about the configuration of the soil only at one instant of time. The availability, first, of dedicated X-ray beams at synchrotron facilities, then the advent of commercially-available table-top X-ray scanners since the turn of the century, raised expectations that via X-ray computed tomography (CT), one could routinely observe aggregates in undisturbed soil samples, and therefore monitor them over time. If earlier depictions of soil aggregates as "levitating", as Garland et al. (2023) adequately describe them (see their illustration of it in Figure 1), or, equivalently, as "surrounded by macropores" (Bouma et al., 1998) had been correct, then a

technique like X-ray CT should have been able to locate aggregates without difficulty. Unfortunately, that was not to be. The experience that many of us have had with efforts in this sense has been very frustrating. Even in CT images of artificial soil samples consisting of repacked "aggregates", it is not straightforward at all to identify the boundaries of the aggregates (e.g., Juyal et al., 2019, 2021).

Thank you for this very thorough history of the challenges associated with identifying soil aggregates. We are familiar with these challenges and are aware of and in fact agree with the importance of analyzing undisturbed soils to better understand and quantify many soil functions. However, as described in the abstract (L22-23: "Here, we illustrate how aggregates can form and dissipate within the context of undisturbed, intact soils, highlighting the point that aggregates do not necessarily need to have a distinct physical boundary and can exist seamlessly embedded in the soil.") and introduction (L84-87: "We do not attempt to choose a side in the "solid phase perspective" versus "pore space perspective" debate, as we believe there is in fact no contradiction between these concepts for describing soil structure. Rather, we aim to demonstrate that there is no incongruity between the existence of aggregates and the fact that we often cannot see them in undisturbed soil, which we hope helps resolve some of the conflicting views, and ultimately advances our understanding of soil functioning."), the aim of our manuscript is simply to establish that aggregates exist naturally embedded in the soil as areas where soil particles are more strongly bound to each other than neighboring particles. We are not arguing that soil aggregates are the same as those isolated using destructive analysis methods, since that is simply not the case (L104-106). The issue of whether or not it is necessary to take aggregates explicitly into account when trying to understand the effect of the architecture of soils on soil functions, while undoubtedly tremendously important, is not the focus of this manuscript.

The title of an article by Koestel et al. (2021) briefly gave hope that a novel technique had been developed to "delineate aggregates in intact soil using X-ray imaging". However, in the body of that article, the authors conceded that they had not yet been able to achieve that goal. In that context, one could argue that the manuscript by Garland et al. (2023) puts a final nail in the coffin of the aggregate perspective, since these authors admit explicitly that, except in surface horizons of arable soils after tillage, "in practice aggregates are often not visible in undisturbed soils or in deeper soil layers at the same spatial scale". They further state that aggregates "may, but do not necessarily need to be, bordered by pore space at spatial scales relevant for most soil mechanistic investigations". In clear, this means that for these authors the definition of aggregates does not call for them to be separated from each other by a distinctive inter-aggregate pore space, i.e., does not require them to have "a distinct physical boundary".

Again, our goal is not to propose that soil aggregates isolated using destructive methods (i.e. "fraggregates") are the same as those naturally created in the soil. It is quite the opposite in fact. Naturally occurring soil aggregates are simply two or more primary soil particles that cohere more strongly to each other than neighboring particles. In most contexts this would not produce a distinct or discrete soil aggregate resembling a "fraggregate". As mentioned above, we added a sentence to make this point more explicit (L104-106). Additionally, we have further emphasized this point by stating (L126-130): "As we have discussed, the term 'soil aggregate' is used to indicate that certain

soil particles cohere to each other more strongly than neighboring particles. It does not give any indication of the size, shape, strength, or general arrangement of the particles and voids that make up that aggregate. In contrast, to these naturally formed, *in-situ* soil aggregates, the size, shape and composition of destructively sampled soil aggregates are often described in relation to the technique used and force applied to isolate them" as well as (L136-138): "While aggregates isolated from soils using such destructive techniques cannot be directly linked to in-situ soil aggregates using current methods, understanding how differences in mineral-mineral and organo-mineral binding sites impact chemical and physical soil characteristics including soil porosity (Fukumasu et al., 2021; Weng et al., 2021) is paramount."

In practice, in line with the day-to-day experience of CT experts and image analysts over the last 20 years, this implies that in general we have no way of knowing whether a given chunk of soil inside an undisturbed soil sample is an aggregate until, at some instant of time, we either resin-impregnate the soil and obtain thin sections in which we can try to delineate the aggregate using micromorphological techniques, or we dismantle the soil sample entirely and isolate the aggregate that way. Logically, the inability to observe aggregates at any other time before that precludes any serious, observation-based research on the temporal evolution of individual aggregates, and in particular on the processes of aggregate formation or aggregate turnover. In this respect, it is worth noting that the cute illustrative example of Garland et al.'s (2023) Figure 4, in which they depict the evolution of soil aggregates over time, could never be confirmed nor refuted experimentally at this stage since the only information one could have access to would be related exclusively to one of the 6 times that are depicted, but not to all 6 of them.  Furthermore, equally problematic is the fact that since we have no way of determining what the boundary conditions of a given aggregate we manage to isolate were in the original, undisturbed soil, it is impossible to even envisage to replicate those conditions *ex situ* to carry further experiments on the aggregate (Kravchenko et al., 2019: Baveye et al, 2022).

We understand that it is quite frustrating to not be able to visualize the dynamic nature of these soil aggregates as well as one would like. Moreover, since aggregates as we discuss here are not separate entities, but rather areas where soil particles are more strongly bound together than others, visually identifying an individual, distinct object completely separated from neighboring soil particles is not realistic. However, the fact that we currently cannot visualize soil aggregates in the same way as soil pores or destructively isolated aggregates (i.e. "fraggregates") does not imply that they do not exist. While we can indeed visualize the heterogeneity of organo-mineral soil particles in a CT image, for example, this unfortunately does not give us any indication of its dynamic nature, as you point out. However, this is very similar to a CT image of soil pores within an intact soil core. While we know that soil pores are dynamic, appearing and disappearing in response to various stimuli, this is not captured from a single X-ray image. Furthermore, even though we can analyze an intact soil core, the core is ultimately still removed from the soil. Even if the soil core is reinstalled into the soil (as it was done in the study highlighted in figure 2), disturbing the soil surrounding the sample is unavoidable. In this sense it is still destructive and does not allow us to follow the evolution of exactly this soil volume because we have already removed the core.

Shortcomings from both the study of soil pores and soil aggregates are indeed frustrating, but part of the nature of studying dynamic processes in an opaque medium.

In spite of the fact that they provide very compelling arguments to abandon the "aggregate perspective" on the effect of soils structure/architecture on soil functions, Garland et al. (2023) still conclude their manuscript by writing that "the question of how aggregates that are seamlessly embedded in soil can be isolated and analysed in more detail *in-situ* to more closely assess microscale processes within aggregates and in relation to pore structures still remain a considerable challenge, yet a worthwhile and crucial research goal". So, even though Garland et al. (2023) argue very convincingly that in many soils we should not expect at all to be able to see aggregates, which are "seamlessly embedded" in the soil matrix, nevertheless they still consider that aggregates are important and that it is crucial that we keep carrying research on them.

Again, we disagree that we are providing arguments to abandon the aggregate perspective. As mentioned in the introduction (L84-85), "we do not attempt to choose a side in this debate, as we believe there is in fact no contradiction between these concepts for describing soil structure." Instead, we are trying to show that soil aggregates can and do exist seamlessly in the soil as areas of more or less strongly bound soil particles. The geochemical differences in these heterogeneous compounds, and how they are bound together in the soil, may have significant impacts on carbon stabilization dynamics as well as nutrient turnover, retention and speciation, all of which ultimately impacts the structure of a soil and the biological communities the soil is able to support. Understanding how the heterogeneity of the organo-mineral matrix impacts carbon and nutrient storage (i.e. 'the solid phase/aggregate perspective), as well as how this material is arranged across a geometric matrix (i.e. the 'pore space/architectural perspective) is integral for better understanding soil structure. However, to make it clear that we are not proposing to focus future studies solely on isolation of soil aggregates, but rather how the physicochemical implications of such soil interactions affect overall soil structure, we have now rephrased this section (which was also pointed out by Reviewer #2) to better reflect this meaning (L221-223): "… the question of how aggregates that are seamlessly embedded in soil can be studied in-situ to better understand the role of soil structure in microscale processes remains a considerable challenge, yet a worthwhile and crucial future research goal." We hope this conveys our message more clearly.

To some extent, this abrupt and somewhat surprising volte-face of Garland et al. (2023) is understandable, given the training that most of us in soil science have received. During our studies, it has been customary for instructors to mention that soils are composed of aggregates. Most soil science textbooks, even when their primary focus is soil physics (e.g., Kutílek and Nielsen, 1994; Jury et al., 1991; Jury and Horton, 2004; Lal and Shukla, 2004; Hillel, 2004, 2013; Radcliffe and Simunek, 2018) or soil microbiology (e.g., Alexander, 1961; Paul and Clark, 1989; Coyne, 1999; Tate, 2020), traditionally have an introductory section on texture and one on aggregates. So, aggregates are part of our collective psyche, and it is therefore probably difficult for some soil scientists to let go of the belief that it is important to involve aggregates when we talk about soil processes and functions. It is in this

context, undoubtedly, that Yudina and Kuzyakov's (2019) emotional plea to "save the face of soil aggregates" should be understood. However, the down side of this psychological reluctance to let go of a familiar concept causes us to rarely if ever ask ourselves the question of what part of what we do would not be possible if we did not involve this concept. **Specifically, are there questions, other than those related to aggregates themselves, that we could not answer if we did not account for the presence of aggregates in soils?**

We would argue that the reason why so many soil science textbooks address texture and aggregates is not due to an emotional connection to this topic, but rather because of the well-established role and importance they play in driving soil processes including soil carbon and nutrient stabilization and turnover (Lehmann et al., 2007; Laub et al., 2023). Understanding how carbon and other key elements are stabilized and transformed at the various micro-scale mineral-mineral and organo-mineral binding sites (i.e. aggregates), and how this impacts carbon stabilization and nutrient availability to biological communities living in the soil is but one of the many questions that can be better addressed by taking soil aggregates into account. Relying solely on the assessment of soil architecture (i.e. through X-ray tomography and similar methods) to capture the mechanical components and strengths of soil particle interactions (i.e. as is often done through geophysical, seismic techniques) would unfortunately not contribute to this goal. Neither would this approach give an indication of ecological legacy (i.e. microbial communities and their spatial arrangement), which is critical for understanding ecological functioning. Ignoring these critical components of soil structure would be in line with arguing that the only difference between a sand and clay particle is the size (because that is all you can see), while ignoring the vital geochemical distinctions they hold and which contribute greatly to a multitude of soil processes.

Critically, we acknowledge that ideally the question of the chemical and structural identity of the carbon and biologically-relevant nutrients would further be assessed within the context of the 3D soil-pore structure, as flows of oxygen and water greatly influence these dynamics. It is precisely this interdependency of both the solid phases and pore spaces that supports our proposal that the scientific divide between the 'aggregate/solid perspective' and the 'architecture/pore perspective' "is not only unnecessary, but is in fact hindering the progression of research in the field of soil science" (L80-81), since naturally both aspects of soil structure need to be taken into account for understanding these diverse and complex processes.

Laub, M., Blagodatsky, S., Van de Broek, M., Schlichenmaier, S., Kunlanit, B., Six, J., Vityakon, P., and Cadisch, G.: SAMM version 1.0: A numerical model for microbial mediated soil aggregate formation, EGUsphere [preprint], https://doi.org/10.5194/egusphere-2023-1414, 2023.

Lehmann, J., Kinyangi, J. & Solomon, D. Organic matter stabilization in soil microaggregates: implications from spatial heterogeneity of organic carbon contents and carbon forms. Biogeochemistry 85, 45–57, https://doi.org/10.1007/s10533-007-9105-3, 2007.

An enlightening hint toward the answer to this crucial query is obtained by reviewing some of the textbooks I just mentioned. Take, for example, Hillel's (2004) popular "Introduction to Environmental Soil Physics" textbook. Pages 78 to 88 dutifully report on the size, shape, breakdown, and stability of aggregates in soils. And then, nothing. The 350 pages that follow cover all kinds of processes in soils,

including soil-plant-water relations, and delve into their mathematical description, without ever invoking aggregates. Likewise, Jury and Horton (2004) devote 2 of the 370 pages of their soil physics textbook to the size and shape of aggregates, then subsequently avoid the topic altogether. A similar situation is found in the soil microbiology textbook of Tate (2020), where the author devotes 4 (out of close to 600 pages) to describe aggregates, and then never mentions them again. In Wall's (2012) edited book on soil ecology and ecosystem services, soil aggregates are mentioned briefly in only two of its 22 chapters. Soil chemistry texts are even more revealing; The topic of soil aggregates is not mentioned at all, even briefly, in the overwhelming majority of them (e.g., Bohn et al., 1985; Sposito, 1989; Sparks, 1989; McBride, 1994; Essington, 2021).

There is no doubt that we are only at the infancy of grasping how the chemical, physical and biological heterogeneity of soil mineral-mineral and organo-mineral complexes changes at the micro-scale within the intact structure of the soil-pore matrix. This difficulty in assessing such a complex arrangement of compounds is doubtlessly why most soil chemical analytical methods require grinding and homogenization of the soil sample at hand. To our knowledge there currently does not exist instrumentation that can accurately quantify the chemical and structural composition of nutrients *in-situ* at a scale relevant for understanding larger-scale ecosystem and soil functions. 3-D methods available for such a characterization at the pore-network scale are just emerging (some of the authors of this commentary are involved with some avantgarde papers on this topic) and remain very labor intensive. This is one of the reasons why most work on soil pore networks do not focus on the micro-scale differences in soil chemistry along these channels and their impacts on nutrient flows and biological communities. However, this does not negate the importance that these complex interactions have on driving these essential functions.

On and on I could go like this, reviewing one textbook after the other, over the last 50 years, and demonstrate that virtually all of their authors manifestly disagree with the view that "the major role of aggregates for a broad range of functions [...] cannot be underestimated" (Yudina and Kuzyakov, 2019). However, perhaps the most illuminating illustration in the literature of the lack of relevance of aggregates to the description of essential processes occurring in soils comes from a recent journal article, which several of the authors of Garland et al. (2023) also co-authored. Indeed, Meurer et al. (2020) provide a very enlightening example of how a model concept initially based on aggregates may eventually turn towards an "architectural" perspective for practical and scientific reasons. These authors' aim was to model the dynamic interactions between soil structure and the storage and turnover of soil organic carbon (SOC). At the beginning of their article, they argue that "the aggregated structure of soil is known to protect SOC from decomposition and, thus, influence the potential for long-term sequestration. In turn, the turnover and storage of SOC affects soil aggregation, physical and hydraulic properties and the productive capacity of soil." They then set out to develop a new computational model of the "dynamic feedbacks between soil organic matter (SOM) storage and soil physical properties (porosity, pore size distribution, bulk density and layer thickness)." In the process of developing this model, they first performed a

sensitivity analysis and also investigated parameter identifiability using a synthetic dataset. The outcome of these analyses was that Meurer et al. (2020) decided to focus on the dynamics of the soil pore space, and subsequently invoked the term of aggregation, not in connection with aggregates as earlier in the text, but solely to refer to the generation of additional pore space in soil associated with the presence of organic matter. Based on empirical observations, they assumed a linear relationship between this aggregation pore volume and the volume of SOM. Thus, Meurer et al. (2020) write, "individual soil aggregates are not considered as explicit entities in this model" and, as Kuka et al. (2007) had earlier, they proposed a strictly pore-based model instead. A similar approach was followed by Falconer et al. (2015), who showed that SOM dynamics was regulated by accessibility determined by the soil pore network, as well as the way organic matter and microorganisms are arranged and can move within the pore space. This did not require any a priori assumption about the protection of organic matter within aggregates.

Thank you for these nice examples of modeling. We understand that there is much work being done recently to improve models of soil carbon turnover. Some researchers prefer a pore-based model, as you mention, while others have indeed found improvement of their models by including soil aggregate fractions (i.e. Laub et al., 2023). As mentioned above, we believe that taking both the solid and pore phases of a soil system are necessary for understanding soil processes.

Laub, M., Blagodatsky, S., Van de Broek, M., Schlichenmaier, S., Kunlanit, B., Six, J., Vityakon, P., and Cadisch, G.: SAMM version 1.0: A numerical model for microbial mediated soil aggregate formation, EGUsphere [preprint], https://doi.org/10.5194/egusphere-2023-1414, 2023.

Thus, if we return to what I asked earlier, namely whether there are questions, other than those related to aggregates themselves, that we could not answer if we did not account for the presence of aggregates in soils, the answer is simple. So far, no one has mentioned a single question that we cannot address because we do not know how to deal with aggregates appropriately, in practice or in theory.  Neither has anyone suggested practical ways to involve aggregates quantitatively in models in order to make the latter adhere more closely to experimental observations. In that context, the admission by Garland et al. (2023) that in general we should not expect to see aggregates in undisturbed soils is very significant and should motivate us to keep doing in the future what the overwhelming majority of our predecessors appear to have done in the past, i.e., not focus on the concept of aggregates to make progress. That is not to say that the alternative, the "architectural perspective", will not be challenging. But at least it involves soil characteristics that we can measure (via CT and other techniques, which are constantly improving), it allows us to carry out laboratory experiments to understand things at a deeper level and, last but certainly not least, it does not require us to deal with entities that we cannot see or whose existence we cannot ascertain without entirely disrupting the soil that allegedly contains them.

Again, thank you for your opinion. We agree that the answer is simple- simply 'yes'. Above we gave multiple examples of questions we could answer when taking

aggregates – and pore spaces – into account for better understanding soil structure controls and dynamics. In our manuscript we do not make a call to "focus on the concept of aggregates to make progress", as you state, and thus it is possible that our main aim or goal was not interpreted as we intended. Indeed, as stated in L22-25: "Here, we illustrate how aggregates can form and dissipate within the context of undisturbed, intact soils, highlighting the point that aggregates do not necessarily need to have a discrete physical boundary and can exist seamlessly embedded in the soil. We hope our contribution helps the debate on soil aggregates and supports the foundation of a shared understanding on the characterization and function of soil structure" and L84-87 "We do not attempt to choose a side in this debate, as we believe there is in fact no contradiction between these concepts for describing soil structure. Rather, we aim to demonstrate that there is no incongruity between the existence of aggregates and the fact that we often cannot see them in undisturbed soil, which we hope helps resolve some of the conflicting views, and ultimately advances our understanding of soil functioning," we are aiming to show how the 'aggregate perspective' and 'pore perspective' are in fact not at odds with each other.

As stated nicely by Reviewer #2:

"Specifically, if, in agreement with the authors, we define aggregates as the localities within the soil matrix where cohesion among adjacent soil particles is stronger and porosity is lower than in their immediate surroundings, then we are talking about natural heterogeneity of densities and coherencies within an intact soil body. Nobody in their right mind, starting with the proponents of the intact-soil view, would argue against the existence of such heterogeneity. In fact, this heterogeneity is what many of us are studying using X-ray CT tools. It is great that the authors made a convincing case that the aggregates (defined as per above) do not need to have strong or visible or, for that matter, any inter-aggregate pore boundaries."

We are trying to show that most likely researchers from both sides of this debate are discussing the same structure, but are speaking about it in different terms and in a different context. If we can agree to having this shared understanding of this basic characterization of soil, we believe that future discussions on this topic will benefit.

**CC2:**

I would like to thank the authors for this contribution to an important debate, although one of their major conclusion is probably that there should not be a debate. As first author of one of the articles that fueled this debate (Vogel et al. 2021) I would like to share a few thoughts on it.

First of all, I completely agree with Garland et al. (2023) that the aggregates we isolate from soil are often seamlessly embedded in the soil matrix which is characterized by heterogeneous binding forces that change dynamically mainly related to the turnover of organic matter as illustrated in their Figure 4. This is about the same message as was intended to be conveyed by Figure 2 of Vogel et al. (2021). The conclusion of Vogel et al. (2021) was, however, that we certainly need to consider the spatial fabric of organic and mineral compounds but in relation to the pore space which cannot be seen as being describable by inter- and intra-aggregate pores – a concept that is also challenged by Garland et al. (2023). This is an essential part of the holistic approach proposed by Vogel et al. (2021) which I think is perfectly consistent with the observation of seamlessly embedded aggregates.

But what are the consequences?  I think Garland et al. (2023) stop at the critical point where the debate ignites. What is the essence of seamlessly embedded aggregates? It is obvious that describing aggregates in terms of size and shape is directly and necessarily related to their boundaries. But what if theses boundaries cannot be defined?  An obvious conclusion is that the birth moment of an "aggregate" is when we isolate it by whatever means from its natural environment. Hence, the term aggregate is potentially misleading since it implies a process of aggregation but in fact it is a process of fragmentation which is the origin its formation. Thus, we should rather call them "fraggregates".  All this refers to the type of "aggregates" that are thought to be produced as illustrated in Figure 4 of Garland et al. (2023) and not to those aggregates where the generation process is well known and which can be clearly identified (e.g. faunal casts, pseudo sands, polyeders in shrinking clay).

Thank you for your much appreciated comments. We agree that aggregates as they are commonly known in the soil science community (i.e. "fraggregates") can only come into being with their typical visible size and shape boundaries during the isolation/fragmentation process. Here we are trying to distinguish between these artificial "fraggregates" and actual aggregates, which are defined as "any group of soil particles that coheres more strongly to each other than neighboring particles" (Martin et al., 1955; SSSA, 1997). To highlight this further, we have now added (L104-106): "As a result, aggregates formed *in-situ* will inherently not look the same as destructively isolated aggregates, but rather appear seamlessly embedded in the heterogeneous organo-mineral matrix, punctuated at various points by pore spaces, as described by Vogel et al. (2021)."

This accurate definition and conceptualization of aggregates, as naturally arising and seamlessly embedded in the soil, seems to somehow have been lost, and replaced

almost entirely by the idea of "fraggregates", which we propose has, at least in part, created such an opposition toward this conceptualization of soil structure (or aggregates). In our opinion, establishing this distinction is the first step in moving the debate, or discussion, on this topic forward. We agree that there is much more to discuss regarding this topic (i.e. what (if any) are the implications of these aggregates on soil functioning and mechanics, how do they influence soil pore occurrence and distribution, etc.). However, our opinion is that if there is better agreement amongst soil scientists regarding the presence of heterogeneously bound soil particles within a soil matrix (i.e. aggregates), then a more open, fruitful, and progressive discussion of these remaining points could be possible.

The conclusion that the term "aggregates" might be misleading is not just quibbling. I am convinced that the wording has a substantial impact on our perception of soil structure formation and functioning. The focus on aggregates in recent research fostered and solidified the notion that aggregates are viewed as self-organized, functional units and the pore space is reduced to inter- and intra-aggregate pores. This was the major critique formulated by Vogel et al. (2021) and this potentially challenges the basic assumptions of some of the recent research approaches on aggregates. This is the reason for the debate.

We agree that terminology is of upmost importance in this discussion. It undoubtedly influences one's opinion on this matter, and we argue that this term, coupled with images of aggregates "floating" in the soil matrix, helps to foster some of the confusion and debate on this issue, as such boundary conditions are simply not realistic. Indeed, perhaps developing another term that more accurately reflects the heterogeneous organo-mineral matrix that essentially is soil would be helpful for developing a shared understanding of the dynamic relationships that we are discussing here.

In the updated manuscript we added the sentence given above (L104-106), which emphasizes that aggregates are formed and exist within the context of the 'organo-mineral matrix', which references the Vogel et al. (2021) article. However, until an alternative term is more formally developed, we wanted to simply and directly highlight that aggregates are not necessarily (nor often) the distinct, self-organized functional units that are often portrayed or produced by destructive isolation methods. Rather, aggregates exist embedded in soils, regardless of our (in)ability to see them with current imaging techniques, and that because of the differences in binding strengths between soil solids, this will add to characteristic physical, chemical, and biological soil properties beyond simply the arrangement of solids and pores.

Remarkably, the final conclusion of Garland et al. (2023) demonstrates how the traditional view of aggregates which is burned into the minds of many soil scientists might direct research in a doubtful direction: "... *the question of how aggregates that are seamlessly embedded in soil can be isolated and analyzed in more detail in-situ to more closely assess microscale processes within aggregates and in relation to pore structures still remains a considerable challenge, yet a worthwhile and crucial future research goal.*" Why don't we allow the thought that the aggregate concept should be replaced by a more appropriate one? For example, the formation of an organo-mineral soil matrix with heterogeneous binding forces permeated by a pore system through which energy and binding agents are provided?

This is an interesting idea, and we thank you for proposing it. Essentially, we agree completely with your description of soils as an "organo-mineral matrix with heterogeneous binding forces permeated by a pore system through which energy and binding agents are provided". The point of our manuscript was to try to highlight the point that soil aggregates are not disparate, self-organized units, but are in fact rather more as you describe. Nonetheless, we do believe that further investigations of this heterogeneity (i.e. especially chemical differences associated with different binding strengths and patterns) are worthwhile for further understanding soil functioning within an in-tact soil. However, as this was also brought up by Reviewer #2, we have now rephrased this sentence by removing the words '*be isolated*' to (L221-223): "… the question of how aggregates that are seamlessly embedded in soil can be studied *in-situ* to better understand the role of soil structure in microscale processes remains a considerable challenge, yet a worthwhile and crucial future research goal." This adaptation removes the emphasis on analyzing "fraggregates", but rather calls for further developing ways to analyze aggregates *in-situ* and in relation to their geometric orientation with pores and other soil solids.

Regarding replacing the aggregate concept with a more appropriate concept, we believe that this would be quite an undertaking that is outside the scope of this particular manuscript. Our approach here was to instead clarify that the 'aggregate concept' is, for all intents and purposes, the same as what you are calling the 'organo-mineral soil matrix' concept. We believe that the understanding of what an aggregate is may be the first hurdle in getting this understanding across.

Overall, I think Garland et al. (2023) and Vogel et al. (2021) have pretty much the same understanding, but I do not agree that this debate is unnecessary, and is even „hindering the progression of research in the field of soil science". It is essential to critically question established concepts for the progress in science and I have the impression that this debate already inspired many colleagues and will support progress in our understanding of soil structure and its functioning.

Thank you for your opinion on this matter. We would like to rephrase our statement by emphasizing that we think that the *divide* that this debate causes is unnecessary, but certainly not the *discussion*. We have now clarified this in the update manuscript by changing "we hope our contribution helps alleviate the debate" to "we hope our contribution helps the debate" (L24).

**RC1:**

In the paper provided, the authors set out to try on two conflicting views on soil structure – solid and pore perspectives. In my opinion, they almost succeeded, as they provided a simple and, most importantly, viable abstraction to describe the delineation of the different functional zones within soil structure (which are actually aggregates) (*Figure 3*). This is the most important and strongest takeaway of this paper. Also, despite the simplicity and brevity of Section 3, I would like to emphasize its value for those readers who may wish to develop the topic of soil structure. Especially, I would like to highlight the validity of the thesis "a more holistic assessment must inevitably consider multiple complementary approaches". I have some suggestions and questions regarding the terminology and the wholeness of the proposed approach, which do not lessen the value of proposed MS. The MS contains the necessary novelty, and is written quite logically. Therefore, it can be accepted after a moderate revision. See my general and specific suggestions below.

Thank you for your overall positive assessment of our manuscript, and for your comments and suggestions to improve and streamline our main message.

**General comments**

Since the authors set out to try on two groups of researchers, it would be nice to see explicitly which of the main theses presented by both sides the authors find useful, and some arguments regarding the non-contradiction (L81) (in a general sense, I agree with you). For example, the authors do not use or discuss the definitions given in previous articles (*Vogel et al., 2022; Yudina and Kuzyakov, 2023*). As Prof. Vogel correctly points out in the preprint comments "*the wording has a significant impact on our perception*".

Thank you for your suggestion. We agree that wording has a significant impact on our perception, and have thus ensured that the terminology used throughout the text is consistent. However, rather than try to reintroduce and discuss differences in terminologies as has already been done recently (i.e. Vogel et al., 2022; Yudina and Kuzyakov, 2023), we added both of the most commonly used terms from these papers to describe both viewpoints (i.e. 'solid phase' and 'aggregate' perspective as well as 'pore space' and 'architecture' perspective) (L78-79 and L218-219). Moreover, our manuscript emphasizes (L203-207) that "despite methodological limitations, continuing to investigate the spatial arrangement of solid particles and pores as well as the flow and transfer of substances throughout the soil pore network and how the solid particles are connected remains critical. The composition and bioavailability of substances bound in aggregates, and how this influences soil microbial communities and soil fauna living within or on the surface of aggregates and pores is paramount to better understanding how soil structure drives functioning." Thus, we are highlighting that characterizing both the solids and pore spaces, as well as their spatial arrangement and chemical, physical, and biological makeup are of upmost importance for better understanding soil structure.

You need to better disclose what you mean by "do not necessarily need to have a distinct physical boundary" (L21) without jeopardizing the use of the term

"aggregates" (which is actually what Prof. Baveye immediately appeals to in his comment to the preprint). In my understanding, this is analogous to applying the concept of soil types at the landscape scale, and trying to identify corresponding to different soils areas. It is also often impossible to distinguish *distinct* physical boundaries between them (as well as between other environmental bodies characterized by gradual transitions). Because many phenomena in nature are continuous, they don't know how to have sharp boundaries. However, we nevertheless use such abstractions and more than successfully. Accordingly, I don't think the word " distinct" is quite appropriate here. Because the distinctness will be achieved due to the conditions that we will set (real, i.e. experimentally affecting the soil, or virtually, i.e. modeling the processes).

Thank you for your opinion. We understand your point and have now changed 'distinct' to 'discrete' in the revised manuscript (L19). You are correct that there will be distinct physical boundaries, related to whatever boundary condition is placed on them. In contrast, 'distinct' indicates an aggregate that is separate from the rest of the soil, which is what we are arguing is not occurring nor possible within in-tact soils. Therefore, the word 'discrete' is better able to represent our meaning since it indicates the individuality of the different aggregates without referring to a distinct visual difference. Thank you.

From L136-138, it is not entirely clear what the authors' position is – whether they consider size, shape, etc. characteristics important for aggregates. Moreover, from L138-139 it appears that the authors take an operational approach – aggregates are what is distinguished in such and such a way. This to some extent contradicts the authors' position that aggregates exist as natural entities (L89), and the aggregate size differs (L141-143, L209). Please clarify your point.

Thank you for bringing this to our attention. With this sentence we were trying to say that this definition of an aggregate does not specify that an aggregate should be any particular size or shape. Instead, the specific process used to isolate soils more strongly bound than neighboring soils will result in soil aggregates of an operational, characteristic size and shape. In our revised manuscript we have clarified this point to avoid this apparent contradiction by stating (L126-130): "As we have discussed, the term 'soil aggregate' is used to indicate that certain soil particles cohere to each other more strongly than neighboring particles. It does not give any indication of the size, shape, strength, or general arrangement of the particles and voids that make up that aggregate. In contrast to these naturally formed, *in-situ* soil aggregates, the size, shape and composition of destructively sampled soil aggregates are often described in relation to the technique used and force applied to isolate them."

I think it will be important to emphasize that the delineation of boundaries between aggregates according to the presented conceptual scheme on *Figure 3* (L108-125) will 1) strongly depend on the available instrumental capabilities (which pores and type of contacts are detectable), 2) and, accordingly, have different significance depending on the type (hierarchy) of the aggregates that the task is to delineate. If we are talking about the experimental application of your approach (that looks promising). For example, at a rather good resolution of tomographic imaging (of the

order of the first μm), we do not see pores, which can actually constitute a meaningful fraction (see Gerke et al., 2021). Accordingly, if one is trying to apply what the authors suggest to research, the question arises as to how it should be used. It would be good to reveal this point to a greater extent within the Section 3 of MS.

Thank you for your thoughts on this matter. We agree that the delineation of boundaries in a practical application will indeed depend on the instrumentation and the scale/research question under investigation. However, it is not our goal in this manuscript to describe in detail precisely how aggregates should be delineated and under what context and conditions. However, we have now added the following sentence to highlight these points (L200-203): "The questions of if, how and at what scale the chemical and physical components of soil aggregates can be assessed *in-situ*, and how this is related to environmentally relevant functions depends on instrumental capabilities and the specific research question at hand (Gerke et al., 2021; Amelung et al., 2023)."

I find the argument against inter- and intra-aggregate pore space (L97-101) as interesting, but in need of refinement. It highlights the weak side of the paper – you do not explicitly discuss the hierarchy of soil structural organization. But actually discussing it (e.g., here L182-185). In my opinion, without this abstraction (as pores belonging to different dynamical parts of the structure), as well as without the concept of non-aggregated mass (which you do not use exept L165 during describing t0 and beyond), presented approach does not look quite complete. Likely such abstractions would have improved the description in the L161-188.

Thank you for your suggestion to strengthen our discussion of the hierarchy of soil structural organization. You are correct that the detailed description of the hierarchy of these aggregate- and pore-forming processes is not complete, as these are very complex and dynamic processes and beyond the scope of the paper. Moreover they have already been reviewed and discussed in detail (Yudina and Kuzyakov, 2023). Therefore, to avoid reiterating a topic that has already been covered, we specify our intention with this paragraph by stating the following (L141-144): "We do not aim to model real processes, nor discuss how the hierarchy of aggregate and pore formation influences their physical and functional properties (Yudina and Kuzyakov, 2023), but instead to show how aggregates may appear and disappear as the binding strengths between soil aggregates and particles strengthen and weaken over time."

**Specific comments**

L9 it's not very correct to write "soils and land-use types" with "and", because they are different kinds of things; land use type has no aggregation, only soils

Good point. We have now removed "land-use types" from this sentence so that it reads "… nearly all soils across the globe (L9)."

L27 what is the difference between solids and aggregates? voids and pores? – It's not clear what the point of listing alternative terms here is. Please, try to use terms more strictly within the whole text of the MS.

We have now removed the word 'voids' from this sentence to avoid listing similar terms. By "solids" here we are referring to solid materials that are part of the soil matrix such as rocks, decomposing roots, and other organic matter debris. While they are not central to our discussion on soil aggregation, they are an important part of soils in general, which is why we wanted to explicitly name them as a component of soil. Rather than take out this word, have added (L29) "(including organic material and stones)" to this sentence to clarify our meaning.

L55 temperate ◊ temperature

Thank you for catching this. We have now changed 'temperate' to 'temperature' in the revised manuscript.

L66 "the unrealistic boundary conditions of isolated aggregates" conditions for what, modelling? or boundary conditions for isolation of aggregates?

By "boundary condition" we are referring to the idea that aggregates *in-situ* need to be bound on all sides by pores in order to be delineated within an intact soil. We have clarified this point by adapting the sentence to (L69-70): "… as well as the unrealistic boundary conditions of isolated aggregates (Kravchenko et al., 2019; Vogel et al., 2021) that are completely separated from surrounding soil particles."

L70 probably, the relevant works where this is said should be cited at the end of this sentence

As was suggested by Reviewer #2, we have now avoided using phrases such as "proof that aggregates do not exist" in the revised version. Therefore, this part of the sentence has been removed, and replaced with (L72-74) "Albeit only very few attempts have been made to corroborate or falsify this claim (e.g. Koestel et al., 2021), here we argue that such *in-situ* identification is not necessary, and in many cases is not realistic, for verifying the existence of soil aggregates."

L95-97 it will be good to put this argument to the abstract, as this is one of the important foundations on which your proposal is built; however, it is not clear for me what you mean here – "at spatial scales relevant for most soil mechanistic investigations". Please, clarify. E.g., microaggregates having size of several-tens-hundreds μm and surrounded by micro- and mesopores are relevant for rheological behavior.

Thank you for your suggestion. We have now added this sentence to the abstract and removed 'at spatial scales relevant for most soil mechanistic investigations'. It now reads "Given that soil aggregates consist of any group of soil particles that coheres more strongly to each other than neighboring particles, aggregates may, but do not necessarily need to be, bordered by pore space" (L20-21).

L98 "if aggregates do not need to be physically separated" ◊ prefer to write "cannot be surrounded on all sides by pores". Because you actually make an excellent point that the physical boundary can be not only the pore, but also various types of contacts between solids.

Thank you for your suggestion. We have now reworded this sentence as you suggest in the revised manuscript (L101-102): "Our viewpoint here challenges the concept of intra-versus inter-aggregate pore space: as aggregates cannot be surrounded on all sides by pores, there is not necessarily a distinctive inter-aggregate pore space."

L109 "the different soil particle fractions (i.e. sand, silt and clay)" ◊ the term "soil particle fractions" is not strict enough – as for the soil texture fractions it would be more accurate to write "soil particle size fractions". If you mean them. According to the bracketed text, you do. However, it would be more logical and consistent with the rest of the text to specify simply «soil solids» instead of this term referring to soil texture rather than soil structure. Please, clarify this.

Thank you for bringing this to our attention. We agree with your point and have now rephrased this part of the text (L109) to 'soil solids', as you suggest, to remain more in-line with the rest of the text.

L138 "Instead ... " – I cannot agree with this sentence in any way, because if aggregates are natural formations of soil, their size and other characteristics are determined by soil processes, but not by methods used.

Your comment is absolutely correct, and we did not mean to imply that the size and characteristics of soil aggregates are created solely by soil aggregate isolation methods. Rather, we were trying to say that if soil aggregates are not visible in-situ (because they are not fully bound by pores, as we discuss), then it is primarily the isolation technique that creates characteristic sizes and shapes reported in most studies on soil aggregate properties. However, we fully agree with you that the natural processes creating soil aggregates are the primary factors regulating the size, shape and chemical properties of the aggregates- we are just arguing here that this will happen/is happening regardless of whether they are visible in-situ or not. We have now added clarification to this point in the revised manuscript to avoid any potential misunderstandings (L126-130).

L152 "Aggregates do not need to be separated by distinct physical boundaries." would be better named –> "Types of physical boundaries between the aggregates". Pores and different types of contacts between soil solids are physical boundaries. Our task as researchers is to define correctly the boundary conditions for different types of aggregates.

Thank you for your suggestion. We have now changed the title of Figure 3 to "Types of physical boundaries between aggregates" (L393), as you recommend.

**RC2:**

I would like to thank the authors for resuming this important conversation, as well as for bringing into question all too commonly seen misleading illustrations of soils as assemblages of disjoint aggregates. The manuscript is well written, and the topic is well covered. So below are not criticisms of the manuscript, but a couple of thoughts that came to my mind while reading it.

We greatly appreciate your overall positive assessment and suggestions for improvement.

I would like to venture stating that the existing "opposition between the viewpoints" does not seem as drastic as suggested by the authors. Both the destructive soil sieving-based aggregate analyses and the intact soil X-ray CT (and such) scanning are just two different types of tools we have at our disposal to study soil structure. Here, as everywhere else in science, for better or worse, the available tools shape the concepts and research directions. For many past decades the wide accessibility of sieving as a tool to study soil structure led to an expansion of soil aggregation concepts relying on enumeration and quantifications gained via sieving. Now the ever-widening accessibility of X-ray CT scanning leads to new types of data and new conceptual (pore-based) frameworks.

In general we agree with you, both regarding how ever-evolving research tools often shape the direction research and new ideas take as well as your opinion that the opposition between viewpoints is likely not as strong as it is often claimed to be. Yet regardless of the strength of this opposition, it is clear that opposition to some degree certainly exists (i.e. Kravchenko et al., 2019; Wang et al., 2019; Yudina and Kuzyakov, 2019). Moreover, while we do not think that opposition is in principle a negative issue (opposition can and often does spark worthwhile and elucidating discussions), we do not think that it is helpful to repeatedly negate other researchers' chosen analytical approach in favor of other options without fully understanding their reasoning. We believe that misinterpretation of the term "aggregate" plays a role in this lack of understanding and apparent conflict, and it is for this reason we are trying to clarify this small, yet important, point. Nevertheless, we understand your point and thus have minimized the reference to "conflicts" of viewpoints on this topic by replacing "… some researchers have questioned whether aggregates exist at all" with "… some researchers have questioned the relevance of aggregates for soil processes" (L64-65).

The "aggregates do not exist" statement seems to be turning into a "strawman" these days. None of the proponents of the pore-based or, rather, intact-soil view of the soil structure ever said that the "aggregates do not exist". In fact, their every publication is abounded with examples of soils and soil horizons with aggregates of pedogenic origin. It is the aggregates of artificial sieved origin that are objectionable. Specifically, if, in agreement with the authors, we define aggregates as the localities within the soil matrix where cohesion among adjacent soil particles is stronger and porosity is lower than in their immediate surroundings, then we are talking about natural heterogeneity of densities and coherencies within an intact soil body. Nobody in their right mind, starting with the proponents of the intact-soil view,

would argue against the existence of such heterogeneity. In fact, this heterogeneity is what many of us are studying using X-ray CT tools. It is great that the authors made a convincing case that the aggregates (defined as per above) do not need to have strong or visible or, for that matter, any inter-aggregate pore boundaries. However, as also noted by the authors, once an intact soil is subjected to sieving it is not at all certain that this heterogenous density and cohesion will be well reflected in the soil fragments generated by sieving. Thus, let's do not grab the sensational "aggregates do not exist" statement, but allow in the rest of the argument. What we will hear then is that the "aggregates" procured through sieving (or as Dr. Vogel aptly named them "fraggregates") are highly unlikely to exist within the soil as unique separate entities.

Thank you for understanding and nicely summarizing one of our main points. This is an incredibly important aspect that we were indeed trying to highlight. As you correctly state, "nobody in their right mind, starting with the proponents of the intact-soil view, would argue against the existence of such heterogeneity." Within this framework, by showing that aggregates are not detached, isolated units "floating" in this matrix, but are simply embedded in this matrix as areas of relatively more strongly bound soil particles, we are hoping that scientists from both sides of the debate can come together with such a shared understanding of an aggregate.

Moreover, in the revised manuscript we changed the sentence from "Albeit only very few attempts have been made to corroborate or falsify this claim (e.g. Koestel et al., 2021), it has been misconstrued as "proof" that aggregates do not exist  to "Albeit only very few attempts have been made to corroborate or falsify this claim (e.g. Koestel et al., 2021), here we argue that such *in-situ* identification is not necessary, and in many cases is not realistic, for verifying the existence of soil aggregates" (L72-74) in order to avoid propagating a "strawman" fallacy. Similarly, in L14, we changed 'the concept of soil aggregates is hotly debated, leading to confusion about their function or even existence' to 'the concept of soil aggregates is hotly debated, leading to confusion about their function or relevancy to soil processes'.

I am not excited by the call for action with which the authors end the manuscript: "the question of how aggregates that are seamlessly embedded in soil can be isolated and analyzed in more detail in-situ to more closely assess microscale processes within aggregates and in relation to pore structures still remains a considerable challenge, yet a worthwhile and crucial future research goal." First, why do we need to isolate something that is seamlessly embedded? Instead of looking at it as a part of the whole? Second, so far, the major tool employed in efforts for such isolation has been soil sieving. And I don't think this tool needs any further encouragement, and this statement might do just that. Sieved soil fragments lost their spatial context, depriving us of the ability to efficiently study soil regions with distinctly different ecological roles. On our sieve there will be fragments originating from rhizosphere, from detritusphere, from bulk soil, etc. Lumping them together and trying to gain information on the role of soil structure in microscale processes is very difficult and highly inefficient, to say the least. Why not, instead, work with intact soil? And to isolate soil micro-samples with specific, e.g., high

density/coherence characteristics from regions of ecological significance, say, rhizosphere or a border of an earthworm channel or bulk soil, etc.? Would this be what the authors have in mind as well?

Thank you for your opinion. We understand the complications with interpreting results from measurements on aggregates isolated from soils using destructive techniques in relation to their role within in-tact soils. In this manuscript we are not suggesting that we should continue struggling with this same dilemma. Instead, by calling for a closer assessment of the micro-scale controls on soil processes arising from aggregates in-situ, we mean that further investigations are needed to better understand and quantify how chemical and physical changes related to the strength and type of binding between soil particles impacts soil functioning. The suggestion you give to isolate "micro-samples" from an intact soils at regions of ecological significance is certainly one possibility, among others. Again, we are argue that many perspectives and approaches are both welcome and needed for adequately assessing soil structure. To address your point that we are not indicating that the study of "fraggregates" should be the focus of future research, we have removed "isolated" from this last sentence (L221-223) so that it now reads: "… the question of how aggregates that are seamlessly embedded in soil can be studied *in-situ* to better understand the role of soil structure in microscale processes remains a considerable challenge, yet a worthwhile and crucial future research goal." We hope this conveys our message more clearly.

I completely agree with the authors that to "investigate the spatial arrangement of solid particles and voids as well as the flow and transfer of substances throughout the soil pore network and how the solid particles are connected remains critical". I just do not understand why do we need "fraggregates" to address this critical need? Can we just skip this middleman and try to look at flows and transfers and their effects on, say, soil organisms inhabiting pore networks directly?

Thank you for your question. There is no question that for many research investigations, it would be perfectly acceptable to "skip the middleman" as you say. However, this is simply not the case for all research questions relating to investigations of soil processes and dynamics. While one could assess the impact of flows and transfers of various substances on soil organisms, others may be interested to better understand how the differences in binding strengths between soil particles and organic matter influences the chemical state and turnover of nutrients stabilized (or not) by these various binding sites, and how these dynamics would influence or contribute to differences in soil microbial community distribution. For example, an accumulation of reactive metal phases (i.e. Al, Mn, Fe) is often found in aggregates (especially microaggregates) (Totsche et al., 2018). These elements accumulate and form geochemically distinct entities and actively contribute to the formation and stability of aggregates, which influences a variety of soil processes since the geochemical composition of this occluded area in soil will not interact and react with its environment in the same way as the soils not enriched in reactive metals. Since the soils and solid particles bordering these pore spaces are composed of a combination of heterogeneous mineral and organic matter particles, such differences and heterogeneity will surely influence nutrient availability and in turn biology at the microscale. Undoubtedly this would be ideal to assess *in-situ*, but until a reliable and accessible methodology is available to study the chemical nature of such binding sites, assessing the chemical nature of different types of "fraggregates" can help reveal many aspects of these critical chemical dynamics in the meantime.

Totsche, K.U., Amelung, W., Gerzabek, M.H., Guggenberger, G., Klumpp, E. Knief, C., Lehndorff, E., Mikutta, R., Peth, S., Prechtel, A., Ray, N., and Kögel-Knabner, I.: Microaggregates in soils, J. Plant Nutr. Soil Sci., 181, 104-136, https://doi.org/10.1002/jpln.201600451, 2018.